# Effects of Caponization on Growth Performance and Carcass Composition of Yangzhou Ganders

**DOI:** 10.3390/ani12111364

**Published:** 2022-05-26

**Authors:** Mingming Lei, Xiaolu Qu, Zichun Dai, Rong Chen, Huanxi Zhu, Zhendan Shi

**Affiliations:** 1Institute of Animal Science, Jiangsu Academy of Agricultural Sciences, Nanjing 210014, China; 20140036@jaas.ac.cn (M.L.); qquuxxiiaaoolluu@163.com (X.Q.); 20210064@jaas.ac.cn (Z.D.); chenrong_big@163.com (R.C.); 2Key Laboratory of Crop and Livestock Integration, Ministry of Agriculture, Nanjing 210014, China

**Keywords:** gander, caponization, live weight, gene expression

## Abstract

**Simple Summary:**

Goose meat is recognized as one of the healthiest foods. Goose capons are specially bred and consumed in several parts of China for their high-quality meat. However, the effects of caponization on goose growth and carcass traits have remained uninvestigated, and its molecular mechanisms remain unclear. In this research, caponization lowered testosterone and increased the total cholesterol and triglyceride concentrations in serum. Caponization increased live weights by promoting food intake and abdominal fat deposition, and improved meat quality by increasing intermuscular fat. Changes in the expression of these genes indicate that caponization increases the live weight mainly by increasing fat deposition rather than muscle growth. These results expand our understanding of the mechanisms of caponization on growth performance and fat deposition in ganders.

**Abstract:**

In this study, we determined the effects of caponization on the growth performance and carcass traits of Yangzhou ganders. Fifty sham operated geese (the control group) and 80 caponized geese (the caponized group) were selected at 150 days of age and reared until 240 days of age. At 210 days of age, 30 geese from the caponized group were selected and fed with testosterone propionate (testosterone group). The results showed that caponization lowered testosterone and increased the total cholesterol and triglyceride concentrations in serum, live weights, average 15 day gains, and feed intake. Abdominal fat and intramuscular fat were significantly higher in the caponized geese than in the control at 240 days. Gene expression analysis showed that caponization promoted abdominal fat deposition and intermuscular fat content by upregulating the expression of adipogenic genes in the liver, adipose tissue, and muscle tissue. The high expression of *SOCS3* in the hypothalamus, liver, and muscle of caponized geese suggests that caponization may lead to negative feedback regulation and leptin resistance. Changes in the expression of these genes, along with the downregulation of *PAX3* in the breast muscle and *MYOG* in the leg muscles, indicate that caponization increases the live weight mainly by increasing fat deposition rather than muscle growth. These results expand our understanding of the mechanisms of caponization on growth performance and fat deposition in ganders.

## 1. Introduction

Testosterone is a sex hormone secreted by the Leydig cells of the testes that stimulate the reproductive system as well as the growth of muscle, bone, and connective tissue [1,2]. In male chickens, testosterone inhibits abdominal fat accumulation and negatively regulates adipogenesis-related genes [3,4] that promote fat synthesis and increase fat accumulation [5,6,7]. Caponization, the surgical removal of the testes, results in testosterone deficiency or a reduction in testosterone levels, which lead to the degeneration of secondary male sexual characteristics such as the comb, wattle, fighting behavior, and vocalizations [8] in addition to higher body weights and more savory meat [9,10,11]. Some studies reported that caponized cocks had increased live weights (LWs) [12,13,14], while others showed that caponization resulted in an overall increase in abdominal fat weight (AFW), as well as subcutaneous and intramuscular fat in chickens [4,15,16]. The accumulation of body fat improves meat quality by enhancing the flavor, texture, and juiciness of the meat compared to intact cocks [8].

Goose rearing is an important form of poultry production, and goose meat consumption has increased in China and several European countries [17]. According to statistics, the total goose production in China is approximately 2.52 million tons, and approximately 95.2% of goose meat is consumed annually [18]. Goose capons are specially bred and consumed in several parts of China for faster growing and better quality meat than intact ganders [19]. However, the effects of caponization on goose growth and carcass traits have remained uninvestigated, and its molecular mechanisms remain unclear. Therefore, this study aims to investigate the effects of caponization on the LW, feed intake (FI), and carcass performance of Yangzhou ganders by comparing them to intact males to understand the role of testosterone in ganders.

## 2. Materials and Methods

### 2.1. Ethics Approval

The experimental procedures were approved by the Research Committee of Jiangsu Academy of Agricultural Sciences and conducted with adherence to the Regulations for the Administration of Affairs Concerning Experimental Animals (Decree No. 63 of the Jiangsu Academy of Agricultural Science on 8 July 2014).

### 2.2. Animal Populations and Experiment Design

The experiments were carried out on Yangzhou ganders (aged 150 d) in Anhui Tianzhijiao Goose Industry Co., Ltd. (Quanjiao, Anhui, China). The geese were foot-marked, weighing 4.75 ± 0.20 kg, and the population density was 4 geese/m^2^ on the ground. The geese were raised in the conventional method of stocking and supplementary feeding (11.69 MJ/kg ME, 12.5% CP, 0.11% calcium, and 0.14% available phosphorus for 150-day-old geese) [20]. In addition to feed, the geese were free to eat grass during grazing. The geese were maintained under natural daylight and temperatures. Fifty sham-operated geese (the control group) and eighty caponized geese were selected at 150 days of age and reared until 240 days of age. Ten geese were randomly selected from each group to be slaughtered at the ages of 180 d and 210 days. At 210 d, sixty caponized geese were divided into two groups: the caponized group and the testosterone group. The geese of the testosterone group were fed testosterone propionate (Sigma-Aldrich, Tokyo, Japan). The dose of testosterone propionate was 5 mg per kg of body weight per day. Testosterone propionate was dissolved in sesame oil and mixed with the feed to feed these geese for 30 d in the testosterone group. The amount of testosterone propionate per kg of feed was 150 mg. In order to ensure that all feed mixed with testosterone propionate was consumed, each subgroup was first fed with about 2 kg mixed testosterone propionate. After eating all the feed, feed without testosterone was added to ensure that all geese could eat freely. Ten slaughters from each group were performed at the age of 240 days. The geese were fasted 16 h before slaughter. Ten geese were randomly selected from each group to be slaughtered at the age of 180 days and 210 days. They were electrically stunned (120 V/50 Hz for 5 s) and exsanguinated by severing the jugular vein and carotid artery on one side of the neck. Afterward, they were passed through a warm scalding vessel (60 °C for 2 min) and a plucker (2 min) and were manually eviscerated. The tissue samples of hypothalamus, pituitary, liver, breast muscle, leg muscle, and abdominal fat were collected and were frozen in liquid nitrogen, and stored at −80 °C.

### 2.3. Testectomy

Caponization was performed at 150 days of age. The testectomy procedure was performed according to Chen et al. [21]. The ganders selected for surgery were deprived of feed and water for 12 h, respectively, before the procedure. Anesthesia was performed using xylazine (Rompun, Bayer HealthCare, Leverkusen, Germany) and ketamine (Ketaset, Fort Dodge Animal Health, Fort Dodge, IA, USA) in doses of 23 mg/kg of body weight. Anesthesia was administrated via goose flipper vein. The (mean ± SE) body weight of the ganders was 4.75 ± 0.20 kg. After the removal of the feathers and the disinfection of the skin with povidone alcohol 75%, a 1.5-cm incision was made between the 2 last ribs. A rib spreader was inserted and the testicles were removed. The control group had a 1.5-cm incision between the last two ribs without removing the testicles. Survival rates for the whole experiment were 100%, respectively.

### 2.4. Measurements

LW of each goose was measured every 15 days individually, and FI of 30 geese per cell was monitored daily. Each group were divided into three subgroups with 10 geese, and the feed intake of each subgroup was measured. The average feed intake of each group was calculated according to the subgroups. Carcass traits were measured according to Symeon et al. [22]. The breast was separated from the back at the shoulder and along the junction of the vertebral and sternal ribs. The legs were separated from the carcass by cutting through the iliofemoral joint and included the thigh and the drumstick. The breast and left leg were dissected into meat, and their wet weights were recorded. AFW was dissected out and weighed. Liver weight and gizzard weight were measured. The intramuscular fat (IMF) was analyzed using the FoodScan Meat Analyzer (Foss, Hillerod, Denmark).

Blood samples were collected via wing veins into heparinized syringes every 15 days. Serum was separated from the blood within 3 h of sample collection by centrifugation at 2000× *g* and stored at −20 °C until the measurements of other targets were conducted. Serum testosterone concentrations were determined by an ELISA using the quantitative Diagnostic Kit for testosterone (North Institute of Biological Technology, Beijing, China) [23]. Concentrations of glucose, triglycerides, cholesterol, high-density lipoprotein, and low-density lipoprotein were measured by using a ROCHE Modular P800 Automatic Biochemical Analyzer (Roche, Milan, Italy) [24].

### 2.5. RNA Isolation, Primer Synthesis, and Quantitative Real-Time PCR

Total RNA from the hypothalamus, pituitary, abdominal fat, breast muscle, and leg muscle was extracted with TRIzol using a commercial kit according to the manufacturer’s instructions (RNAiso Plus, Code No. 9108; Takara, Shiga, Japan). Gene-specific primers were designed using Primer 3.0 software accessd on 8 January 2022 (www.ncbi.nlm.nih.gov/tools/primer-blast/) based on the GenBank databases. ABI PRISM_7500 sequence detection system (Applied Biosystems, Foster City, CA, USA) was used to detect the amplification products. The relative expression levels of different genes in the tissues were calculated according to the 2^−ΔΔCT^ method.

### 2.6. Statistical Analysis

All values are expressed as mean ± SEM. Differences in the plasma concentrations of testosterone, metabolite concentrations data, LW, average weight gain, carcass composition weights, and gene expression levels were analyzed by 2-way ANOVA in the animal experiment with the time and group treatments as factors using SPSS 20 software (SPSS Inc., Chicago, IL, USA). Least significant difference pairwise comparisons were also analyzed for time course and testosterone propionate. Caponization effect on FI was analyzed by ANCOVA, with the cumulative FI of each time period as dependent variable and the total FI of the whole day as the independent co-variable. Statistical significance was set at *p* < 0.05. All pictures were drawn with the GraphPad Prism V8.0 (GraphPad Software, San Diego, CA, USA).

## 3. Results

### 3.1. FI and Growth Performance

FI were higher in the caponized group than in the control group throughout the experiment (Figure 1A). From 210 days to 240 days, the FI of the caponized group was >50 g higher than that of the control group, and even more than >180 g/d at 238 d. The FI of the testosterone group decreased compared to the caponized group, but was higher than that of the control group. During the period from210 days to 240 days, the FI of the of the caponized group was significantly higher than that of the control group (*p* < 0.05).

During the course of the experiment, the LWs of caponized geese became significantly higher than those of the control group at 195 d (Figure 1B, *p* < 0.05). The difference gradually decreased (*p* > 0.05) but reached a significant level again at 240 d (*p* < 0.05). At the end of the experiment, the LW of caponized geese was 6.63 kg, which was 0.6 kg higher than that of the control group.

During the course of the experiment, the weight gains of the control group and caponized group increased, reached the highest at 195 d, and then decreased (Figure 1C). From 225 days to 240 days, the weight gains of the control group decreased sharply, resulting in them being significantly lower than those of the caponized group (Figure 1C, *p* < 0.05). The weight gains of the control group were lower than those of the caponized group from 150 days to 165 days (*p* < 0.05), from 180 d to 195 d (*p* < 0.05), from 195 days to 210 days (*p* < 0.05), and from 225 days to 240 days (*p* < 0.05).

### 3.2. Measure of Plasma Testosterone and Metabolite Concentrations

In the control group, the circulating concentrations of testosterone increased between 150 days to 225 days and subsequently decreased slightly from 225 days to 240 days (Figure 2A). The concentrations in the caponized group remained stable at a low level. A significant difference was observed in the testosterone concentration between these two groups (*p* < 0.05). After the geese were supplemented with testosterone, the serum testosterone concentration of the geese was significantly higher than that in the caponized group and significantly lower than that in the control group (*p* < 0.05).

The glucose of the control group was maintained at approximately 10 mm/mol throughout the experiment. Although the glucose of the caponized group changed slightly, the difference was not significant (Figure 2B, *p* > 0.05). The total cholesterol exhibited an upward trend in the control and caponized groups, and was higher in the caponized group than in the control group (Figure 2C, *p* < 0.05). After supplementation with testosterone propionate, total cholesterol decreased slightly compared to the caponized group and was higher than that of the control group (Figure 2C). Triglycerides of the caponized group were significantly higher than those of the control group at 180 days and 225 days (Figure 2D, *p* < 0.05). After supplementation with testosterone propionate, there was no significant difference in triglyceride concentration between the testosterone group and the caponized group. The high-density lipoprotein concentration of the caponized group was higher than that of the control group at 180 days (Figure 2E, *p* < 0.05). In contrast, the low-density lipoprotein concentration of the caponized group was lower than that of the control group, and the difference was significant at 180 days (Figure 2F, *p* < 0.05).

### 3.3. Carcass Composition

The carcass composition of birds is presented in Table 1. The gizzard weights (GZWs) of the caponized group increased quickly and were significantly higher than those of the control group at 180 days, 210 days, and 240 days (*p* < 0.05). The ratio of GZW to body weight (BW) in the caponized group was significantly higher than that of the control group at 180 days and 210 days (*p* < 0.05), and there was no difference at 240 days. There were significant differences in the liver weights (LIWs) and the ratios of LIW to BW between the control group and the caponized group at 180 days (*p* < 0.05). The AFW increased continuously in all the geese in our experiment. There were significant differences in the AFWs between the caponized geese and control ganders at 210 days (*p* < 0.05) and at 240 days (*p* < 0.01), respectively. After the caponized geese were fed testosterone, their AFW decreased significantly; however, it was still significantly higher than that of the control group. The ratios of AFW to BW in the caponized group and testosterone group were significantly higher than those of the control group at 240 days (*p* < 0.05). There were no significant differences in the breast muscle weights (BMWs) and the ratios of BMWs to BW among the three groups at 180 days, 210 days, and 240 days (*p* > 0.05). The leg muscle weights (LMWs) of the caponized geese were significantly lower than those of the control group at 180 days (*p* < 0.05), and there were no differences at 210 days and 240 days. The ratios of LMW to BW in the caponized group and testosterone group were significantly lower than those of the control group at 240 days (*p* < 0.05). At 240 d, the IMF contents of the breast muscle and leg muscle in the caponized group were significantly higher than those in the control group and testosterone group (*p* < 0.05), and there was no significant difference between the control and testosterone group.

### 3.4. Gene Expression

#### 3.4.1. Appetite Regulating Genes in the Hypothalamus

Both the genes *AgRP* and *NPY* were upregulated in the hypothalamus of the caponized group compared to the control group, and significant differences were observed in the expression levels of the *POMC* gene at 240 days (Figure 3, *p* < 0.05). After supplementation with testosterone propionate, there was no significant difference in the expression of the *AgRP*, *NPY*, *POMC*, and *MC4R* genes among between testosterone group and caponized group, and between testosterone group and control groups (*p* > 0.05).

#### 3.4.2. Expression of Genes in the Liver and Abdominal Adipose Tissues

To gain further insights into the effect of caponization on adipogenesis metabolism, 12 genes in the liver were examined (*LEPR*, *AR*, *INSR*, *SOCS3, PCK1*, *SCD*, *SQLE*, *RXRG*, *PPARγ*, *FASN*, *FABP4*, and *ACC*). Caponization significantly upregulated the expression of the genes *SOCS3*, *SCD*, *SQLE*, *RXRG*, and *FASN* in the liver (*p* < 0.05), and the testosterone propionate supplementation downregulated the expression of *SOCS3*, *SCD*, *SQLE*, *RXRG*, and *FASN* (Figure 4, *p* < 0.05). These 12 genes were also detected in the abdominal adipose tissues. The expression of the genes *SOCS3*, *PCK1*, *SCD*, *RXRG*, *FASN*, and *FABP4* was significantly upregulated in the abdominal adipose tissues of the caponized geese (Figure 5, *p* < 0.05), whereas the expression of the gene *ACC* was significantly downregulated (*p* < 0.05). The testosterone propionate supplementation decreased the expression of the genes *SOCS3*, *FASN*, and *FABP4* significantly compared to that in the caponized geese (*p* < 0.05).

#### 3.4.3. Expression of Genes in the Breast and Leg Muscle Tissues

To gain further insights into the effect of caponization on intermuscular adipogenesis, four genes (*FASN*, *FABP4*, *MYOG*, and *PAX3*) were examined in the leg and breast muscles (Figure 6). Caponization significantly upregulated the expression of the genes *FASN* and *FABP4* in the leg muscle tissues (*p* < 0.05), and the testosterone propionate supplementation significantly downregulated the expression of the genes *FABP4* and *FASN* (Figure 6A, *p* < 0.05). The expression of the gene *MYOG* in the leg muscle tissues of the caponized geese was downregulated compared to that in the control geese (Figure 6B, *p* < 0.05).

Four genes (*FASN*, *FABP4*, *MYOG*, and *PAX3*) were examined in the breast muscle tissues. The expression of *FASN* and *FABP4* were significantly upregulated in the breast muscle tissues of the caponized geese (Figure 6C, *p* < 0.05), and the testosterone propionate supplementation resulted in the downregulation of the expression of *FASN* compared to that in the caponized geese (Figure 6D, *p* < 0.05). The expression of the *PAX3* gene in the breast muscle tissues of the caponized geese was downregulated compared to that of the control geese (Figure 6D, *p* < 0.05).

## 4. Discussion

The results of this study showed that caponization reduced testosterone and significantly increased the concentrations of serum cholesterol and triglycerides in geese. Additionally, it enhanced the LW and FI of caponized geese. Caponization also improved the AFW and intermuscular fat content by upregulating the expression of the adipogenic genes in liver tissues, AFW, and muscle tissues. Supplementation with testosterone propionate decreased the LWs and FI of caponized geese.

Caponization promoted the LW of the caponized geese in our study. Moreover, the weight gain of the caponized geese was mainly due to fat deposition rather than muscle growth. The total cholesterol serum concentration increased significantly in the caponized group compared to the control group during the whole experiment; serum triacylglycerol concentration increased significantly in the caponized group compared to the control group at 180 d and 225 d; in particular, the AFW and the ratio of AFW to BW were higher in the caponized geese than in the control group at 240 d. These results showed that caponization resulted in a reduction in testosterone levels and increased fat deposition. In an earlier report, capons showed a higher total cholesterol concentration than intact males in blood constituents [25], indicating that lipid synthesis increased in capons compared to intact males [26,27]. Lower serum testosterone levels caused by caponization also depressed lipase and enzymes related to fat metabolism, thereby increasing the total cholesterol and triacylglycerol levels in the serum [28,29,30]. The increase in AFW suggested that adipogenesis was significantly faster in the caponized geese than in the control group. In fact, the expression of adipogenic genes, such as *RXRG* and *FASN*, was significantly upregulated in the liver and adipose tissue of caponized geese. In this regard, it has been reported that *FASN* gene expression in liver may be associated with hepatic steatosis [31]. IMF content is an important factor affecting meat quality and nutrition [32]. Caponization also increased the IMF contents of muscle tissues in our study. At 240 d, the IMF contents of the breast and leg muscles in the caponized geese were significantly higher than those in the control geese. The expression of *FABP4* gene related with IMF was upregulated in the breast and leg muscle tissue of the caponized geese compared to the control geese. IMF positively influences flavor, juiciness, tenderness, firmness, and the overall acceptability of the meat [33]. Thus, capons have more savory meat compared to intact ganders.

Testosterone plays an important role in the regulation of muscle growth in males [30,34,35]. High testosterone levels promote muscle growth [36], whereas low testosterone levels decrease muscle growth [30,37]. The results of the present study showed that the LMWs of the caponized group were lower than those of the control group at 180 d. The ratios of LMW to BW in the caponized group and testosterone group were significantly lower than those of the control group at 240 d. This indicated that the muscle growth in the caponized geese was slower than that in intact ganders, which was consistent with the expression of genes related to muscle growth. The expression of the *MYOG* gene in the leg muscle tissues and the expression of the *PAX3* gene in the breast muscle tissues of caponized geese were downregulated compared to the control geese. *MYOG* can promote the terminal differentiation of single free myoblasts and their fusion to form multinucleated myotubes, thereby promoting muscle growth [38,39]. The *PAX3* gene is important at the onset of myogenesis [40,41]. Lower serum testosterone levels decreased the muscle growth of the caponized geese. Therefore, caponization resulted in a reduced level or lack of testosterone, which in turn reduced muscle growth.

Caponization leads to low testosterone levels, and the decrease in testosterone results in an increase in leptin levels in castrated rats [42]. A significantly negative relationship between serum leptin and testosterone has been reported [42]. A high leptin concentration suppresses the JAK/STAT pathway in leptin receptor signaling by negative regulators such as the suppression of cytokine signaling 3 (SOCS3) gene (leptin resistance) [43]. Src homology 2 (SH2) domains are the key domains of SOCS3 that interact with proteins phosphorylated on the amino acid residue tyrosine [44]. Androgen receptor tyrosine phosphorylation may also modulate its ability to interact with the SH2 domains of cell signaling molecules [45]. In the present study, the expression of the *SOCS3* gene in the caponized geese was significantly higher than that in the control group in the hypothalamus, liver, and abdominal adipose tissues. Such a high concentration of *SOCS3* suggested a state of ‘leptin resistance’ in the caponized geese. Leptin resistance stimulated appetite, and the expression of the orexigenic genes, *NPY* and *AgRP*, was upregulated in the hypothalamus, and that of the anorexigenic genes, *MC4R* and *POMC*, was downregulated. Consequently, FI and fat deposition increased in caponized geese. The results were consistent with the observation that castrated mice fed high-energy diets promotes fat deposition [46]. The results demonstrate the importance of testosterone and leptin in the regulation of FI in ganders.

## 5. Conclusions

In conclusion, caponization led to low or a lack of testosterone, and increased FI by upregulating the expression of *NPY* and *AgRP* in hypothalamus. Caponization promoted fat deposition and intermuscular fat by upregulating the expression of adipogenic genes in the liver, adipose tissue, and muscle tissue, and increased body weight and intermuscular fat content. Although caponization promoting weight gain in geese may be related to low levels of testosterone or leptin resistance, the specific molecular mechanism is not clear and needs to be further studied.

## Figures and Tables

**Figure 1 animals-12-01364-f001:**
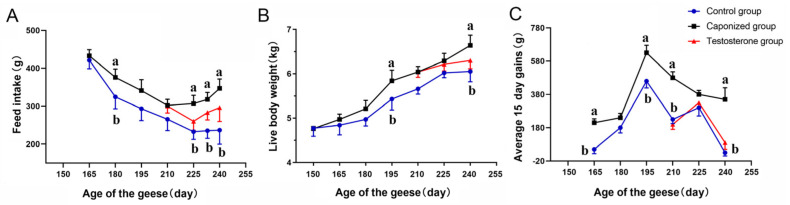
Feed intake (**A**), live body weight (**B**), and average 15 day gains (**C**) of the control group (●), the caponized group (■), and the testosterone group (▲). Data are shown as mean values ± standard error of the mean. Different letters above the bars denote significant differences (*p* < 0.05).

**Figure 2 animals-12-01364-f002:**
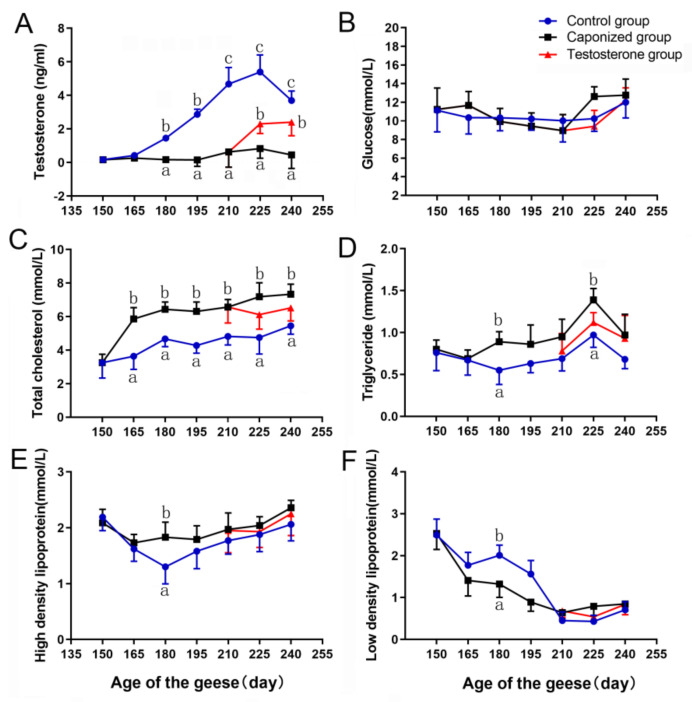
Serum testosterone and metabolite concentrations of the control group (●), the caponized group (■), and the testosterone group (▲). (**A**) (Testosterone), (**B**) (glucose), (**C**) (triglycerides), (**D**) (cholesterol), (**E**) (high−density lipoprotein), (**F**) (low−density lipoprotein). Data are shown as mean values ± standard error of the mean. Different letters above the bars denote significant differences (a,b: *p* < 0.05; b,c: *p* < 0.05; a–c: *p* < 0.01).

**Figure 3 animals-12-01364-f003:**
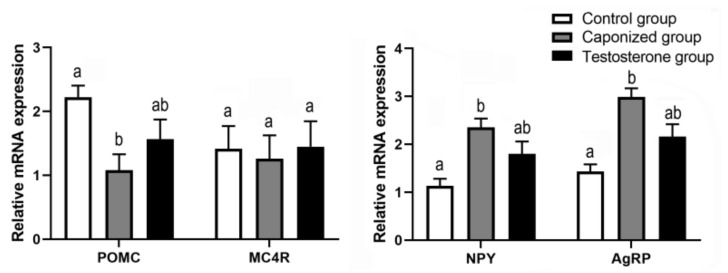
Genes’ mRNA levels relative to *β-actin*, *POMC*, *MC4R*, *NPY*, and *AgRP* of hypothalamus in the control, the caponized group, and the testosterone group. Data are shown as mean values ± standard error of the mean. Vertical bars represent the standard errors of the mean. Different letters above the bars denote significant differences (*p* < 0.05).

**Figure 4 animals-12-01364-f004:**
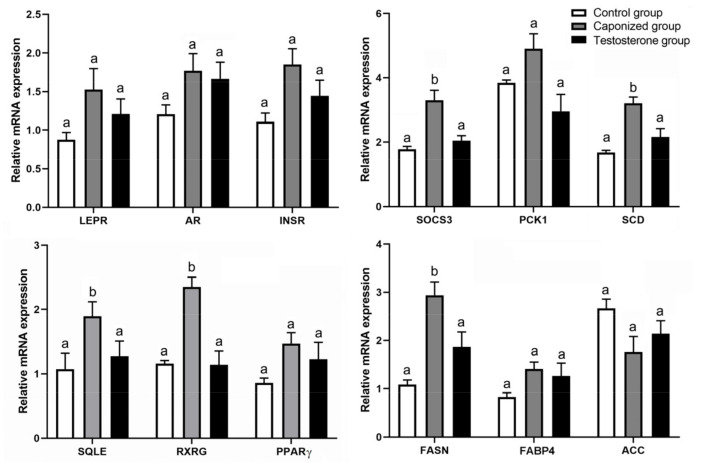
Genes’ mRNA levels relative to *β-actin*, *LEPR*, *AR*, *INSR*, *SOCS3*, *PCK1*, *SCD*, *SQLE*, *RXRG*, *PPARγ*, *FASN*, *FABP4*, and *ACC* of liver tissues in the control, the caponized group, and the testosterone group. Data are shown as mean values ± standard error of the mean. Vertical bars represent the standard errors of the mean. Different letters above the bars denote significant differences (*p* < 0.05).

**Figure 5 animals-12-01364-f005:**
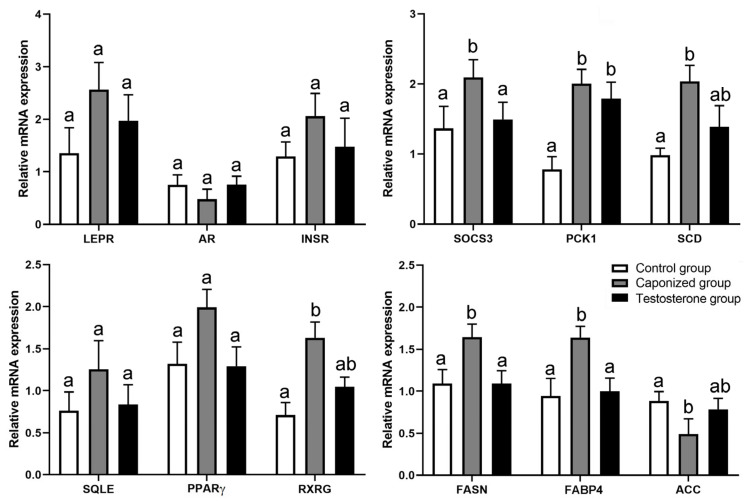
Genes’ mRNA levels relative to *β-actin*, *LEPR*, *AR*, *INSR*, *SOCS3*, *PCK1*, *SCD*, *SQLE*, *PPARγ*, *RXRG*, *FASN*, *FABP4*, and *ACC* of abdominal adipose tissues in the control, the caponized group, and the testosterone group. Data are shown as mean values ± standard error of the mean. Vertical bars represent the standard errors of the mean. Different letters above the bars denote significant differences (*p* < 0.05).

**Figure 6 animals-12-01364-f006:**
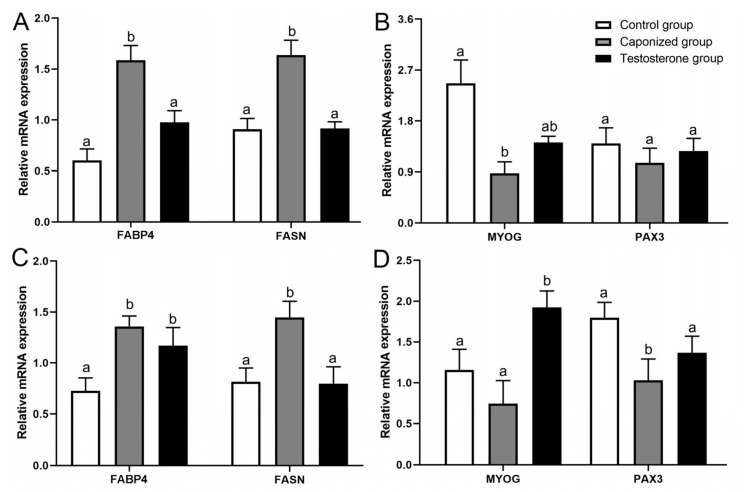
Genes’ mRNA levels relative to *β-actin*, *FASN*, *FABP4*, *MYOG*, and *PAX3* of the leg muscle tissues (**A**,**B**) and breast muscle tissues (**C**,**D**) in the control, the caponized group, and the testosterone group. Data are shown as mean values ± standard error of the mean. Vertical bars represent the standard errors of the mean. Different letters above the bars denote significant differences (*p* < 0.05).

**Table 1 animals-12-01364-t001:** Carcass traits in the control group, the caponized group, and the testosterone group at 180 days, 210 days, and 240 days.

Group	Traits	At 180 Days	At 210 Days	At 240 Days	Ratio of BW ^8^ (%)		
					At 180 Days	At 210 Days	At 240 Days
Control		112.32 ± 4.59 a	106.06 ± 3.93 a	106.90 ± 2.59 a	2.24 ± 0.12 a	1.96 ± 0.09 a	1.78 ± 0.06
Caponized	GZW(g) ^1^	144.53 ± 6.24 c	134.54 ± 6.85 b	127.33 ± 3.55 b	2.86 ± 0.15 c	2.38 ± 0.13 b	1.93 ± 0.06
Testosterone				117.87 ± 7.51 ab			1.84 ± 0.10
Control		85.90 ± 3.87 a	87.66 ± 6.18	100.24 ± 17.01	1.70 ± 0.08 a	1.62 ± 0.13	1.70 ± 0.33
Caponized	LIW(g) ^2^	127.69 ± 18.86 b	87.72 ± 3.92	84.72 ± 6.74	2.56 ± 0.42 b	1.54 ± 0.05	1.27 ± 0.07
Testosterone				100.79 ± 8.35			1.57 ± 0.11
Control		154.79 ± 16.55	178.61 ± 10.37 a	243.69 ± 24.92 a	3.02 ± 0.28	3.26 ± 0.14	3.96 ± 0.30 a
Caponized	AFW(g) ^3^	160.95 ± 17.20	222.19 ± 25.84 b	397.94 ± 32.69 c	3.13 ± 0.29	3.86 ± 0.05	5.95 ± 0.33 c
Testosterone				322.05 ± 30.50 b			4.99 ± 0.40 ac
Control		406.47 ± 22.39	480.70 ± 18.37	544.44 ± 28.62	11.32 ± 0.42	12.37 ± 0.41	12.60 ± 0.31
Caponized	BMW(g) ^4^	377.64 ± 22.04	465.03 ± 15.63	530.45 ± 17.03	11.03 ± 0.37	11.93 ± 0.59	11.85 ± 0.31
Testosterone				563.81 ± 17.13			12.32 ± 0.26
Control		452.54 ± 15.93 a	515.78 ± 12.24	595.50 ± 17.31	12.57 ± 0.22	13.29 ± 0.33	13.76 ± 0.43 a
Caponized	LMW(g) ^5^	415.85 ± 10.81 b	494.00 ± 15.58	553.91 ± 22.59	12.20 ± 0.16	12.60 ± 0.35	12.24 ± 0.31 b
Testosterone				583.23 ± 16.90			11.86 ± 0.28 b
Control		4.59 ± 0.42	4.64 ± 0.38	4.88 ± 0.33 b			
Caponized	IMFB(%) ^6^	4.11 ± 0.33	4.89 ± 0.31	5.84 ± 0.54 a			
Testosterone				4.59 ± 0.27 b			
Control		4.67 ± 0.42	5.34 ± 1.02	4.60 ± 0.38 a			
Caponized	IMFL(%) ^7^	5.07 ± 0.46	6.23 ± 0.76	7.27 ± 0.46 b			
Testosterone				5.34 ± 0.07 a			

^1^ GZW = gizzard weight, ^2^ LIW = liver weight, ^3^ AFW = abdominal fat weight, ^4^ BMW = breast muscle weight, ^5^ LMW = leg muscle weight, ^6^ IMFB = intermuscular fat content of breast muscle, ^7^ IMFL = intermuscular fat content of leg muscle, ^8^ BW = body weight. Different letters above the bars denote significant (a,b: *p* < 0.05; b,c: *p* < 0.05; a,c: *p* < 0.01) differences.

## Data Availability

All data generated or analyzed during this study are included in this published paper.

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
