# Peer review of "Effects of Caponization on Growth Performance and Carcass Composition of Yangzhou Ganders"

_animals, 2022, doi:10.3390/ani12111364_

Round 1
Reviewer 1 Report
Dear editor
My decision: MAJOR revision. Pls see the attached file.
REGARDS

Author Response
- INTRODUCTION does not make enough references to the literature on the subject. Introduction should be extended.
Response: Thanks. We have extended the Introduction, followed as: Goose rearing is an important form of poultry production, and geese meat consumption has been increased in China and several European countries [17]. According to statistics, the total goose production in China was approximately 2.52 million tons and approximately 95.2% of geese meat consumed annually [18]. Goose capons are specially bred and consumed in several parts of China for better quality meat and faster grow than intact gander [19]. However, the effects of caponization on goose growth and carcass traits have remained uninvestigated, and their molecular mechanisms remain unclear. Therefore, this study aims to investigate the effects of caponization on LW, feed intake (FI), and carcass performance of Yangzhou ganders by comparing them to intact males to understand the role of testosterone in ganders.
- If the ethics committee decision was taken in 2014, why was it waited 8 years for publication?
Response: Our experiment started a few months ago, not eight years ago. The Regulations for the Administration of Affairs Concerning Experimental Animals was taken in 2014. Our experiment was approved by the Research Committee of Jiangsu Academy of Agricultural Sciences, and the Committee demanded that we strictly conducted with adherence to the Regulations.
- Which nutrient recommendations wereused?
The ganders were fed ad libitum with mixed feed of crude protein, supplemented with green grass whenever possible, and always had free access to drinking water. …………
Response: We have added the nutrient recommendations, followed as: The geese were raised in the conventional method of stocking and supplementary feeding (11.69 MJ/kg ME, 12.5% CP, 0.11% calcium, and 0.14% available phosphorus for 150-day-old geese) [12]. In addition to feed, the geese were free to eat grass during grazing. The geese were maintained under natural daylight and temperatures.
4. To be corrected as below:
2.4. Measurements Symeon et al. [18]
Response: Thanks. We have corrected it.
5. Give relevant references
2.4. Measurements
Blood samples were collected via wing veins into heparinized syringes every 15 days. Serum was separated from the blood within 3 h of sample collection by centrifugation at 2000 × g, and stored at −20 °C until the measurements of other targets were conducted. Serum testosterone concentrations were determined by an ELISA using the quantitative Diagnostic Kit for testosterone (North Institute of Biological Technology, Beijing, China). Concentrations of glucose, triglycerides, cholesterol, high-density lipoprotein, and low-density lipoprotein were measured by using a ROCHE Modular P800 Automatic Bio-chemical Analyzer.
Response: Thanks. We have added two references in this section: Blood samples were collected via wing veins into heparinized syringes every 15 days. Serum was separated from the blood within 3 h of sample collection by centrifugation at 2000 × g, and stored at −20 °C until the measurements of other targets were conducted. Serum testosterone concentrations were determined by an ELISA using the quantitative Diagnostic Kit for testosterone (North Institute of Biological Technology, Beijing, China) [23]. Concentrations of glucose, triglycerides, cholesterol, high-density lipoprotein, and low-density lipoprotein were measured by using a ROCHE Modular P800 Automatic Bio-chemical Analyzer [24].
6. Results are sufficient.
Response: Thanks.
- Discussion is not sufficient. More in-depth discussion is required. Discussion section is not enough to properly explain reasons of the results
Response: Thanks. We have revised the discussion, and further discussed the effect of intramuscular fat content on the meat quality of caponized geese.

Reviewer 2 Report
Lines 29-30 “Thirty sham-operated geese (the control group) and 40 caponized geese (the caponized group)” In Materials and Methods they are 50 and 80 respectively.
Line 31 “At 210 days of age, 10 geese from the caponized group” In Materials and Methods they are 30.
Lines 103-104 “and FI was monitored daily per cell” how many geese were present in the cell? reports the number.
Lines 149-153 “The total cholesterol and triglycerides exhibited an upward trend in the control and caponized groups, and were higher in the caponized group than in the control group (Fig. 1C, p < 0.05). After the geese were fed testosterone, their total cholesterol and triglyceride levels were higher than those of the control group and lower than those of the caponized group (Fig. 1D)” The upward trend is not evident for triglycerides (especially for the control group). Make two separate comments: one for cholesterol and one for triglycerides.
Lines 197-198 change “The breast muscle weights (BMWs) in the caponized group were lower than those in the control group at 240 d, but there was no significant difference (p > 0.05)” as “The breast muscle weights (BMWs) in the caponized group were lower than those in the control group at 180, 210 and 240 d, but there was no significant difference (p > 0.05)”
Lines 218-220 “The geese that were fed testosterone exhibited downregulated the expression of the genes AgRP and NPY and upregulated POMC and MC4R expression, and there was no significant (p > 0.05). Change exhibited with trend.
Line 326 delete and.
Line 294 after caponized geese. add “In this regard, it has been reported (reference) that FASN gene expression in liver induced may be associated with hepatic steatosis.
Reference -
Liu, X.; Liu, Y.; Cheng, H.; Deng, Y.; Xiong X.; Qu, X. Comparison of performance, fatty acid composition, enzymes and gene expression between overfed Xupu geese with large and small liver. It. J. Anim. Sci. 2020, 20, 102-111 https://doi.org/10.1080/1828051X.2021.1872423
Author Response
Lines 29-30 “Thirty sham-operated geese (the control group) and 40 caponized geese (the caponized group)” In Materials and Methods they are 50 and 80 respectively.
Line 31 “At 210 days of age, 10 geese from the caponized group” In Materials and Methods they are 30.
Response: Thanks. We have revised them, followed as: At 210 days of age, 30 geese from the caponized group.
Lines 103-104 “and FI was monitored daily per cell” how many geese were present in the cell? reports the number.
Lines 149-153 “The total cholesterol and triglycerides exhibited an upward trend in the control and caponized groups, and were higher in the caponized group than in the control group (Fig. 1C, p < 0.05). After the geese were fed testosterone, their total cholesterol and triglyceride levels were higher than those of the control group and lower than those of the caponized group (Fig. 1D)” The upward trend is not evident for triglycerides (especially for the control group). Make two separate comments: one for cholesterol and one for triglycerides.
Response: Thanks. We have revised them, followed as: The total cholesterol exhibited an upward trend in the control and caponized groups, and were higher in the caponized group than in the control group (Fig. 2C, p < 0.05). After supplementation with testosterone propionate, total cholesterol decreased slightly compared to the caponized group and was higher than those of the control group (Fig. 2C). Triglycerides of the caponized group were significantly higher than that of the control group at 180 d, and 225 d (Fig. 2D, p < 0.05). After supplementation with testosterone propionate, there was no significant difference in triglyceride concentration between testosterone group and control group, testosterone group and castration group.
Lines 197-198 change “The breast muscle weights (BMWs) in the caponized group were lower than those in the control group at 240 d, but there was no significant difference (p > 0.05)” as “The breast muscle weights (BMWs) in the caponized group were lower than those in the control group at 180, 210 and 240 d, but there was no significant difference (p > 0.05)”
Response: Thanks. We have revised them.
Lines 218-220 “The geese that were fed testosterone exhibited downregulated the expression of the genes AgRP and NPY and upregulated POMC and MC4R expression, and there was no significant (p > 0.05). Change exhibited with trend.
Response: Thanks. We have revised them, followed as: Both the genes AgRP and NPY were upregulated in the hypothalamus of the caponized group compared to the control group and significant differences were observed in the expression levels of the POMC gene at 240 d (Fig. 3, p < 0.05). After supplementation with testosterone propionate, there was no significant difference in the expression of AgRP, NPY, POMC, and MC4R genes between testosterone group and caponized group, testosterone group and control group (p > 0.05).
Line 326 delete and.
Response: Thanks. We have delete and.
Line 294 after caponized geese. add “In this regard, it has been reported (reference) that FASN gene expression in liver induced may be associated with hepatic steatosis.
Reference -
Liu, X.; Liu, Y.; Cheng, H.; Deng, Y.; Xiong X.; Qu, X. Comparison of performance, fatty acid composition, enzymes and gene expression between overfed Xupu geese with large and small liver. It. J. Anim. Sci. 2020, 20, 102-111 https://doi.org/10.1080/1828051X.2021.1872423
Response: Thanks. We have revised them.

Reviewer 3 Report
The objective of the work was to evaluate the effects of caponization and caponization with testosterone supplementation on growth performance and carcass composition of geese. The work is interesting and apparently does not duplicate information in the literature. There are however, a number of issues that the authors need to address.
In general, the authors need to be careful making statements that gene expression or tissue weight was lower or higher, when the values are not statistically different between groups. In many cases, they values are numerically different, but if they are not statistically different, you should not make the statement or report the change as a trend if the P value is < 0.10. This is a major issue with the paper. If the mean separation shows “a” vs “b”, then they are different. If it is “b” vs “ab” they are not different.
Line 29 The numbers of geese in the Abstract (30 and 40) differ from that reported in the Methods (50 and 80) section (line 81).
Line 60-68 There are no references cited to support the statements made in this section.
Line 76 Can the authors cite a reference that provides the characteristics of this line of geese, their nutrient requirements, etc?
Line 79 Can you cite a reference or detail the composition of the diet used and ensure that the diet met the nutrient requirements of the birds. It is impossible for a reader to repeat these observations without knowing the composition of the diet and how the feed was provided.
Line 82 How were the 10 birds slaughtered on days 180 and 210 selected? Randomly? How were the birds euthanized?
Line 85 Clarify whether the dose of testosterone was per kg of feed or per kg of body weight. Clarify how the birds were fed. Feed intake is reported in Figure 2, but there is no indication how this data was collected. On line 78, it is stated that birds were housed at 4 / meter - how was intake determined in group housed birds?
Line 89 How were the birds euthanized?
Line 96 How were the anesthetics administered? Clarify what was done to the sham group.
Line 133 Again, clarify how FI data was collected. If these are group penned animals, clarify this point.
Line 138 Results. I suggest reporting the body weight and feed intake results (Figure 2) before the blood results (Figure 1).
Line 167 Clarify that you mean 185 g/d higher than the control. Perhaps the 185.29 should be shown as > 180 g/d
Line 184 Figure 2 A - clarify if the feed intake is grams per day. Intake should be singular.
Line 190 Since body weight was greatly increased it would be helpful to express liver, gizzard, muscle and fat pad weights on a relative basis (as a % of BW).
Line 201 and 208 The text here and data in Table 1 reports intramuscular fat for leg and breast muscle, but the procedure used to determine this is not in the Methods section.
Line 219 According to the statistics shown in Figure 3, the POMC, NPY and AgRP expression in testosterone treated is not statistically different than in the caponized group. While they numerically different, you should not state that they “exhibited downregulated” expression.
Line 220 This is an example where the text implies a difference in gene expression that is not statistically different.
Line 223 The Figure 3 legend is not for hypothalamic gene expression.
Line 297 BMW was not significantly different at 180, 210 or 240 days - so you cannot state that the weights were “lower”. In contrast, the LMW was significantly lower.
Line 325 Leptin gene expression was not changed (Figure 4), so this statement is not supported by the data and should be removed. Furthermore, while determination of gene expression is of interest and commonly done, a measure of leptin protein concentrations would be more valuable.
Author Response
The objective of the work was to evaluate the effects of caponization and caponization with testosterone supplementation on growth performance and carcass composition of geese. The work is interesting and apparently does not duplicate information in the literature. There are however, a number of issues that the authors need to address.
In general, the authors need to be careful making statements that gene expression or tissue weight was lower or higher, when the values are not statistically different between groups. In many cases, they values are numerically different, but if they are not statistically different, you should not make the statement or report the change as a trend if the P value is < 0.10. This is a major issue with the paper. If the mean separation shows “a” vs “b”, then they are different. If it is “b” vs “ab” they are not different.
Line 29 The numbers of geese in the Abstract (30 and 40) differ from that reported in the Methods (50 and 80) section (line 81).
Line 60-68 There are no references cited to support the statements made in this section.
Line 76 Can the authors cite a reference that provides the characteristics of this line of geese, their nutrient requirements, etc? Line 79 Can you cite a reference or detail the composition of the diet used and ensure that the diet met the nutrient requirements of the birds. It is impossible for a reader to repeat these observations without knowing the composition of the diet and how the feed was provided.
Line 82 How were the 10 birds slaughtered on days 180 and 210 selected? Randomly? How were the birds euthanized?
Response: Thanks. Ten geese were randomly selected from each group to be slaughtered at the ages of 180 d and 210 d. They were electrically stunned (120 V/50 Hz for 5 s) and exsanguinated by severing the jugular vein and carotid artery on one side of the neck. Afterward, they were passed through a warm scalding vessel (60°C for 2 min) and a plucker (2 min) and were manually eviscerated.
Line 85 Clarify whether the dose of testosterone was per kg of feed or per kg of body weight. Clarify how the birds were fed. Feed intake is reported in Figure 2, but there is no indication how this data was collected. On line 78, it is stated that birds were housed at 4 / meter - how was intake determined in group housed birds?
Response: Thanks. We have revised them, followed as: The dose of testosterone propionate was 5 mg per kg of body weight. Testosterone propionate was dissolved in sesame oil and mixed with the feed to feed 30 caponized geese for 30 d in the testosterone group.
In our previous study, we found that the population density was four geese per square meter, and geese had the best growth performance and reproductive performance. Therefore, we set the population density of each group as four geese per square meter. Since geese cannot be caged, the feed intake of a single goose cannot be measured. So, in each group, we divided 10 geese into a subgroup. The average feed intake of 10 ganders in each subgroup was measured, and the average feed intake of each group was calculated according to the subgroups.
Line 89 How were the birds euthanized?
Response: Thanks. We have added. They were electrically stunned (120 V/50 Hz for 5 s) and exsanguinated by severing the jugular vein and carotid artery on one side of the neck. Afterward, they were passed through a warm scalding vessel (60°C for 2 min) and a plucker (2 min) and were manually eviscerated.
Line 96 How were the anesthetics administered? Clarify what was done to the sham group.
Response: Thanks. Anesthesia was administrated via goose flipper vein. The (mean ± SE) body weight of the ganders was 4.75 ± 0.20 kg. After the removal of the feathers and the disinfection of the skin with povidone alcohol 75%, a 1.5-cm incision was made between the 2 last ribs. A rib spreader was inserted and the testicles were removed. The control group only cut 1.5-cm incision between the last two ribs without removing the testicles. Survival rates for the whole experiment were 100 %, respectively.
Line 133 Again, clarify how FI data was collected. If these are group penned animals, clarify this point.
Response: Thanks. We have revised it in Method. Because geese cannot be raised in a single cage. So, in each group, we divided 10 geese into a subgroup. The average feed intake of 10 ganders in each subgroup was measured, and the average feed intake of each group was calculated according to the subgroups.
Line 138 Results. I suggest reporting the body weight and feed intake results (Figure 2) before the blood results (Figure 1).
Response: Thanks. We have revised it according to the reviewer.
Line 167 Clarify that you mean 185 g/d higher than the control. Perhaps the 185.29 should be shown as > 180 g/d
Response: Thanks. We have revised it, followed as: From 210 d to 240 d, the FI of the caponized group was > 50 g higher than that of the control group, and even more than > 180 g/d at 238 d.
Line 184 Figure 2 A - clarify if the feed intake is grams per day. Intake should be singular.
Response: Thanks. the feed intake is grams per day. We have changed intakes with intake.
Line 190 Since body weight was greatly increased it would be helpful to express liver, gizzard, muscle and fat pad weights on a relative basis (as a % of BW).
Response: Thanks. We have added the data of liver, gizzard, muscle and fat pad weights on a relative basis in Table 1. The data are described in the results, followed as: Carcass composition of birds is presented in Table 1. The gizzard weights (GZW) of the caponized group increased quickly and were significantly higher than those of the control group at 180 d, 210 d, and 240 d (p < 0.05). The ratio of GZW to body weight (BW) in the caponized group were significantly higher than those of the control group at 180 d, 210 d (p < 0.05), and there was no different at 240 d. There were significant different in liver weights (LIWs) and the ratio of LIWs to BW between the control group and the caponized group at 180 d (p < 0.05). The AFW increased continuously in all the geese in our experiment. There were significant differences in AFW between caponized geese and control ganders at 210 d (p < 0.05) and at 240 d (p < 0.01), respectively. After the caponized geese were fed testosterone, their AFW decreased significantly; however, it was still significantly higher than that of the control group. The ratio of AFW to BW in the caponized group and testosterone group were significantly higher than those of the control group at 240 d (p < 0.05). There was no significant difference in the breast muscle weights (BMWs) and the ratio of BMWs to BW among 3 group at 180 d, 210 d, and 240 d (p > 0.05). Leg muscle weights (LMWs) of caponized geese was significantly lower than those of the control group at 180 d (p < 0.05), and there were no different at 210 d and 240 d. The ratio of LMWs to BW in the caponized group and testosterone group were significantly lower than those of the control group at 240 d (p < 0.05). At 240 d, IMF of breast muscle and leg muscle in the caponized group were significantly higher than that in the control group and testosterone group (p < 0.05), and there was no significant between the control and testosterone group.
Line 201 and 208 The text here and data in Table 1 reports intramuscular fat for leg and breast muscle, but the procedure used to determine this is not in the Methods section.
Response: Thanks. We I have added the measurement method of intramuscular fat for leg and breast muscle to the materials and methods, followed as: The intramuscular fat was analyzed using the FoodScan Meat Analyzer.
Line 219 According to the statistics shown in Figure 3, the POMC, NPY and AgRP expression in testosterone treated is not statistically different than in the caponized group. While they numerically different, you should not state that they “exhibited downregulated” expression.
Line 220 This is an example where the text implies a difference in gene expression that is not statistically different.
Response: Thanks. We have revised them, followed as: After supplementation with testosterone propionate, there was no significant difference in the expression of AgRP, NPY, POMC, and MC4R genes between testosterone group and caponized group.
Line 223 The Figure 3 legend is not for hypothalamic gene expression.
Response: Thanks. We have revised them.
Line 297 BMW was not significantly different at 180, 210 or 240 days - so you cannot state that the weights were “lower”. In contrast, the LMW was significantly lower.
Response: We have revised it, followed as: The results of the present study showed that LMWs of the caponized group were lower than those of the control group at 180 d, 210 d, and 240 d.
Line 325 Leptin gene expression was not changed (Figure 4), so this statement is not supported by the data and should be removed. Furthermore, while determination of gene expression is of interest and commonly done, a measure of leptin protein concentrations would be more valuable.
Response: Thanks. Because the sequence of goose leptin gene has not been reported. Therefore, in this experiment, we studied the expression of leptin receptor and androgen receptor,but there were no different.
Due to the low level of blood leptin in poultry, it is difficult to measure the concentration of leptin. In 2008, Hen et al. established a cell receptor signal detection system that can be activated by leptin. The system successfully detected the concentration of leptin in the serum of humans, dairy cows and other mammals and Xenopus laevis, and successfully detected the concentration of leptin in the serum of quail, layer chicken, broiler chicken, and turkey. In 2018, we published an article in collaboration with the Friedman-Einat team. In that study, we made the antibody into freeze-dried powder and sent it to Israel. However, due to the restrictions of customs and transportation conditions, it is difficult to send blood to Israel, so we can't measure the concentration of leptin in blood. Now we are establishing a new method to detect the serum leptin concentration of poultry, hoping to be successful as soon as possible.

Round 2
Reviewer 1 Report
All recommendations were done.
REGARDS
Author Response
Thanks.
Reviewer 3 Report
The authors have responded to my concerns. However, there are a few more details needed.
I appreciate that they have explained the method for determining feed intake. However, if there were 10 birds in a cage, then the experimental unit is the pen or cage, not the individual. It is not valid to calculate intake per bird and have the same statistical power as with individual body weights of the birds. Based on the Methods, there were 3 pens of caponized and 3 pens of testosterone treated. Therefore, n=3 for feed intake. The Y-axis for figure 1 A should be grams per day.
Line 91 The dose method for administering testosterone is not clear. They state 5 mg/kg of body weight daily, but the birds were group penned and some birds may have consumed more feed or less feed. In order for someone to repeat this study, the amount of testosterone in mg per kg of FEED should be stated. Also, according to Figure 1A, feed intake per day decreased from 400 to 200 g (per day ?). If so, was the concentration of testosterone added to the feed increased to maintain the dose?
Author Response
I appreciate that they have explained the method for determining feed intake. However, if there were 10 birds in a cage, then the experimental unit is the pen or cage, not the individual. It is not valid to calculate intake per bird and have the same statistical power as with individual body weights of the birds. Based on the Methods, there were 3 pens of caponized and 3 pens of testosterone treated. Therefore, n=3 for feed intake. The Y-axis for figure 1 A should be grams per day.
Response: Thanks very much for your suggestion, and it was nice. Since it was impossible to measure the individual feed intake of each goose, the feed intake per pen should be calculated, not the individual. We have revised figure 1 A.
Figure 1. Feed intake, live weight, and average 15-day gain of the control group (●), the caponized group (■), and the testosterone group (▲). Data are shown as mean values ± standard error of the mean. Different letters above the bars denote significant (a,b: P < 0.05) differences.
Line 91 The dose method for administering testosterone is not clear. They state 5 mg/kg of body weight daily, but the birds were group penned and some birds may have consumed more feed or less feed. In order for someone to repeat this study, the amount of testosterone in mg per kg of FEED should be stated. Also, according to Figure 1A, feed intake per day decreased from 400 to 200 g (per day ?). If so, was the concentration of testosterone added to the feed increased to maintain the dose?
Response: Thanks. Because geese cannot be raised in a single cage and are highly stressed, they cannot be injected with testosterone propionate every day. So, testosterone propionate only was mixed in the feed. The amount of testosterone in per kg of feed was 150 mg. Feed intake is generally related to animal weight, and those who were heavier may be eat more. In order to ensure that all feed mixed with testosterone was consumed, each subgroup was first fed with about 2kg mixed testosterone propionate. After eating all the feed, we added testosterone free feed to ensure that all geese can eat freely.
We have revised it, followed as: The dose of testosterone propionate was 5 mg per kg of body weight per day. Testosterone propionate was dissolved in sesame oil and mixed with the feed to feed these geese for 30 d in the testosterone group. The amount of testosterone propionate in per kg of feed was 150 mg. In order to ensure that all feed mixed with testosterone was consumed, each subgroup was first fed with about 2kg feed with testosterone propionate. After eating all the feed, we added feed without testosterone propionate free feed to ensure that all geese can eat freely.
